# Comparative Anti-Inflammatory Effects of *Salix* Cortex Extracts and Acetylsalicylic Acid in SARS-CoV-2 Peptide and LPS-Activated Human In Vitro Systems

**DOI:** 10.3390/ijms22136766

**Published:** 2021-06-23

**Authors:** Nguyen Phan Khoi Le, Corinna Herz, João Victor Dutra Gomes, Nadja Förster, Kyriaki Antoniadou, Verena Karolin Mittermeier-Kleßinger, Inga Mewis, Corinna Dawid, Christian Ulrichs, Evelyn Lamy

**Affiliations:** 1Molecular Preventive Medicine, University Medical Center and Faculty of Medicine, University of Freiburg, 79108 Freiburg, Germany; phan.khoi.nguyen.le@uniklinik-freiburg.de (N.P.K.L.); corinna.herz@uniklinik-freiburg.de (C.H.); dutra.joaovictor@gmail.com (J.V.D.G.); 2Division Urban Plant Ecophysiology, Humboldt-Universität zu Berlin, 14195 Berlin, Germany; nadja.foerster@hu-berlin.de (N.F.); inga@entomology.de (I.M.); christian.ulrichs@hu-berlin.de (C.U.); 3Food Chemistry and Molecular Sensory Science, Technical University of Munich, Lise-Meitner-Str. 34, 85354 Freising, Germany; kyriaki.antoniadou@tum.de (K.A.); verena.mittermeier@tum.de (V.K.M.-K.); corinna.dawid@tum.de (C.D.)

**Keywords:** *Salix* species, willow bark, acetylsalicylic acid (ASA), SARS-CoV-2 peptides, in vitro, anti-inflammatory effects, PGE_2_, cytokine

## Abstract

The usefulness of anti-inflammatory drugs as an adjunct therapy to improve outcomes in COVID-19 patients is intensely discussed in this paper. Willow bark (*Salix* cortex) has been used for centuries to relieve pain, inflammation, and fever. Its main active ingredient, salicin, is metabolized in the human body into salicylic acid, the precursor of the commonly used pain drug acetylsalicylic acid (ASA). Here, we report on the in vitro anti-inflammatory efficacy of two methanolic *Salix* extracts, standardized to phenolic compounds, in comparison to ASA in the context of a SARS-CoV-2 peptide challenge. Using SARS-CoV-2 peptide/IL-1β- or LPS-activated human PBMCs and an inflammatory intestinal Caco-2/HT29-MTX co-culture, *Salix* extracts, and ASA concentration-dependently suppressed prostaglandin E2 (PGE_2_), a principal mediator of inflammation. The inhibition of COX-2 enzyme activity, but not protein expression was observed for ASA and one *Salix* extract. In activated PBMCs, the suppression of relevant cytokines (i.e., IL-6, IL-1β, and IL-10) was seen for both *Salix* extracts. The anti-inflammatory capacity of *Salix* extracts was still retained after transepithelial passage and liver cell metabolism in an advanced co-culture model system consisting of intestinal Caco-2/HT29-MTX cells and differentiated hepatocyte-like HepaRG cells. Taken together, our in vitro data suggest that *Salix* extracts might present an additional anti-inflammatory treatment option in the context of SARS-CoV-2 peptides challenge; however, more confirmatory data are needed.

## 1. Introduction

Coronavirus Disease 2019 (COVID-19), a newly emerged respiratory infectious disease, is caused by a mutational RNA virus of the beta-coronavirus family, named Severe Acute Respiratory Syndrome Coronavirus 2 (SARS-CoV-2) [1]. After the first case was reported in December 2019 in Wuhan (China), it quickly spread all over the world and was officially accepted as a global pandemic by the World Health Organization (WHO) in March 2020. Since the end of April 2021, this pandemic has already resulted in significantly increased morbidity and mortality worldwide: more than 140 million confirmed cases and more than 3.0 million deaths have been reported so far in the context of COVID-19 [2]. Bacterial co-infections are often found in viral respiratory diseases and present one of the critical causes for morbidity and mortality [3].

Thus far, various studies have mentioned the significant association of secondary bacterial infections with a worse outcome in COVID-19 patients. Among 191 patients, 50% of dead cases had secondary bacterial infections [4]. In a systematic review and meta-analysis, Lansbury and colleagues reported that 7% of hospitalized COVID-19 patients (*n* = 3834) suffered from bacterial co-infection, and intensive care unit (ICU) patients had a higher prevalence than patients in mixed ward/ICU settings [5]. Another meta-analysis also reported that bacterial infection occurred in 6.9% of COVID-19 patients, ranging from 5.9% in hospitalized patients to 8.1% in critically ill patients [6].

In addition to vaccine development, there are currently great efforts from scientists to discover effective treatment options counteracting COVID-19 symptoms. Non-steroidal anti-inflammatory drugs (NSAIDs) are very widely used to alleviate fever, pain, and inflammation, which are common symptoms of SARS-CoV-2 infected patients. The underlying key mechanism of action for acetyl salicylic acid (ASA) and other NSAIDs is the inhibition of pathogen/cytokine-triggered cyclooxygenases (COXs), especially of the COX-2 enzyme and subsequent prostaglandin E2 (PGE_2_). According to a new study from 2021, NSAIDs, including ibuprofen and meloxicam, could influence COVID-19 outcomes by reducing the inflammatory response and antibody production [7].

The multifaceted functions of PGE_2_ in inflammation and immune surveillance have been well-established [8,9]. Many studies have described that PGE_2_ plays a modulatory role in inflammation and immunity in various viral infections, such as herpes simplex virus, rotavirus, coxsackie virus, influenza A virus, respiratory syncytial virus, or hepatitis B virus [10]. Recently, Smeitink et al. hypothesized that PGE_2_ could play a substantial role in COVID-19 pathophysiology in terms of a hyperinflammatory and deregulated immune response. They highly recommended measuring the PGE_2_ level in COVID-19 patients and proposed PGE_2_ inhibition as a promising treatment strategy for preventing patients from severe disease progression and death [11].

In this study, we aimed at evaluating the in vitro anti-inflammatory efficiency of *Salix* cortex extracts in comparison to ASA in the context of SARS-CoV-2 infection. Willow (*Salix* species) bark extracts have been known for centuries to be anti-inflammatory, antipyretic, and analgesic agents [12]. They are widely recognized throughout Europe and are used for conditions related to inflammatory conditions and flu symptoms, including fever and generalized pain [13].

One of the main chemical active ingredients made from willow bark is salicin, which is metabolized in the human body into salicylic acid, the precursor of ASA; however, a complex mixture of main polyphenols is thought to contribute to the overall effect of *Salix* extracts [14,15]. Since COVID-19 patients also experience gastrointestinal symptoms, particularly abdominal pain and inflammatory diarrhea [16], the anti-inflammatory potential was investigated here using, in addition to activated primary human peripheral blood mononuclear cells (PBMCs), also an intestinal Caco-2/HT29-MTX co-culture model.

## 2. Results

### 2.1. Anti-Inflammatory Activity of Salix Extracts and ASA on PGE_2_ Production in Activated PBMCs

In SARS-CoV-2 infected cells, the expression level of pro-IL-1β was seen to be upregulated [17]. Thus, in this in vitro model system, PBMCs were stimulated with SARS-CoV-2 peptide mixture together with IL-1β to trigger PGE_2_ production, a principal mediator of inflammation. Upon activation, the PGE_2_ concentration in PBMCs was 177.6 ± 111.2 pg/mL, which was 155% compared to untreated control cells (Figure 1A). While both tested *Salix* extracts and ASA could block PGE_2_ release concentration-dependently, this was in the order ASA > extract S6 > extract B. Using LPS from Gram-negative bacteria for PGE_2_ stimulation, the PGE_2_ concentration in PBMCs was 2767 ± 1009 pg/mL, and a similar order in PGE_2_ inhibition potential was observed (Figure 1B).

### 2.2. Chemical Analysis of Salix Extracts

Even though the extracts have been standardized to their total phenolic content, as given in Table 1, qualitative and quantitative differences were evident in the salicylates, flavonoids, flavan-3-ols, and others. One pronounced difference between both extracts was seen in the content of 2′-*O*-acetylsalicortin, which was absent in extract B, while at 7.31 mg/mL, was the most abundant compound in extract S6.

### 2.3. Inhibition of COX1/2 Enzyme Activity by Salix Extracts and ASA

The COX enzyme system presents the major pathway catalyzing the conversion of arachidonic acid into prostaglandins, including PGE_2_ [18]. While the COX-1 enzyme is constitutively expressed in all human tissues and necessary for the maintenance of physiological functions, COX-2 enzyme responses to pro-inflammatory stimuli. Thus, the effects of *Salix* extracts on COX-1 (Figure 2A) and COX-2 (Figure 2B) enzyme activities were further investigated. Both extract S6 and ASA, but not extract B, showed a concentration-dependent inhibition on COX-1 and COX-2 enzyme activity.

At the same concentration, extract S6 showed a higher inhibition on COX enzyme than ASA. Interestingly, 2′-*O*-acetylsalicortin, the most abundant compound in extract S6, had neglectable effects on the COX enzyme activity. Thus, other undiscovered compounds in extract S6 might trigger bioactivity. We further studied the effect of *Salix* extracts and ASA on COX-2 protein expression but could not find the expression inhibition in activated PBMCs in any of the tested settings (Figure 2C).

### 2.4. Effect of Salix Extracts and ASA on Cytokine Release from SARS-CoV-2 Peptide/IL-1β Stimulated PBMCs

To further characterize the anti-inflammatory properties observed for the *Salix* extracts and ASA, we next quantified the release of relevant cytokines from activated PBMCs. The baseline levels of cytokines in non-activated cells were between 0 and 7.99 pg/mL. Upon activation with SARS-CoV-2 peptide and IL-1β, the mean concentration of the cytokines IL-1β, IL-6, TNF-α, and IL-10 in PBMCs were 2774 pg/mL (95% CI, 1028–5109 pg/mL), 608 pg/mL (95% CI, 183–2617 pg/mL), 182 pg/mL (95% CI, 110–330 pg/mL), and 37 pg/mL (95% CI, 12–82 pg/mL), respectively. Pre-treatment with *Salix* extracts or ASA resulted in a decreased level of the pro-inflammatory cytokines IL-1β (Figure 3A) and IL-6 (Figure 3B).

Extract S6 in particular, strongly inhibited both cytokines, whereas ASA showed only weak effects. In contrast, extract S6 significantly increased the level of TNF-α at≥ 5 µg/mL (Figure 3C). The anti-inflammatory cytokine IL-10 can both impede pathogen clearance and ameliorate immunopathology [19]. As a unique feature of the cytokine storm in SARS-CoV-2 infection, a dramatic elevation of IL-10 was reported [20], and thus we also investigated the effect of cell treatment on IL-10. As given in Figure 3D, both *Salix* extracts blocked IL-10 release (extract S6 > extract B), while there was no effect of ASA at the tested concentrations.

### 2.5. Anti-Inflammatory Activity of Salix Extracts and ASA on PGE_2_ Release from Activated Caco-2/HT29-MTX Co-Culture

Using a mixture of different pro-inflammatory stimulators on differentiated Caco-2/HT29-MTX (90:10) co-cultured cells, we aimed to simulate intestinal inflammatory processes, as described in a previous study [21]. As shown in Figure 4, significant inhibition of PGE_2_ release was observed with the *Salix* extracts at ≥5 µg/mL (extract S6 > extract B), whereas this was already at 1 µg/mL with ASA in both the supernatant from the apical (Figure 4A) and basolateral (Figure 4B) sides of activated Caco-2/HT29-MTX co-cultured cells.

### 2.6. Anti-Inflammatory Activity of Salix Extracts and ASA on PGE_2_ Production in Activated PBMCs after Intestinal Absorption and Liver Cell Metabolism

Next, an in vitro Caco-2/HT29-MTX (90:10) model was employed that mimicked both enterocytes and goblet cells of the small intestine, together with hepatocyte-like HepaRG cells. Using this model, we aimed to simulate the process of oral drug absorption and metabolism [22] before assessing the anti-inflammatory activity of *Salix* extracts and ASA in activated PBMCs again. A schematic of the in vitro model is given in Figure 5A. As shown in Figure 5B, the inhibition of PGE_2_ release by *Salix* extracts and ASA was still evident after intestinal absorption and liver metabolism.

## 3. Discussion

PGE_2_, a key pro-inflammatory lipid mediator generated by the arachidonic acid pathway, was demonstrated to modulate inflammation and the immune system in the course of various viral infections, such as herpes simplex virus, rotavirus, coxsackie virus, influenza A virus, and hepatitis B virus [10]. A recent clinical study reported significantly elevated PGE_2_ levels in urine samples of COVID-19 patients in comparison with healthy individuals (170 ± 40 ng/mL vs. 18.8 ± 3.8 ng/mL, *p* < 0.01). The reduction of elevated PGE_2_ production was then considered as a potential treatment option for COVID-19 patients [23].

Based on this finding, Smeitink et al. hypothesized that PGE_2_ could play an important role in COVID-19 pathophysiology in terms of a hyperinflammatory and dysregulated immune response [11]. In addition to the primary infection with SARS-CoV-2, patients are also threatened by co-infections, especially from bacteria. Without a doubt, these co-infections are one of the mortality reasons of COVID-19 [4,5,6,24], and the most common bacterial pathogens detected in patients were Gram-negative bacteria [25].

In our study, both *Salix* extracts and ASA showed strong anti-inflammatory effects in terms of suppressing the PGE_2_ release in human PBMCs, activated by either SARS-COV-2 peptide/IL-1β or LPS from Gram-negative bacteria. This concurs with a previous report showing the anti-inflammatory activity of the ethanolic *Salix* extract 1520L in terms of inhibiting the release of TNF-α and PGE_2_ in LPS-activated primary human monocytes (IC50 = 47 µg/mL) [26]. Compared to our study, it is not surprising that this reported efficacy was much weaker because our extract had been standardized to the total phenolic content.

Low doses (typically 75–81 mg/day) of ASA are well-known to irreversibly acetylate serine 530 of COX-1 enzyme, which causes the antithrombotic effect in platelets. At higher doses (650 mg to 4 g/day), COX-2 is additionally blocked in addition to COX-1, which is required to convert arachidonic acid into PGs, subsequently resulting in antipyretic, analgesic, and anti-inflammatory effects [27,28]. In a retrospective, observational cohort study of 412 hospitalized COVID-19 patients, the early use of low-dose ASA (the median dose of 81 mg) resulted in a reduced risk of mechanical ventilation by 44%, ICU admission by 43%, and in-hospital mortality by 47% [29].

Another study demonstrated that decreased mortality in mechanically ventilated COVID-19 patients resulted from systemic anticoagulation [30]. Thus, ASA could be advantageous due to its antiplatelet properties as an inhibitor of COX-1, which subsequently reduces platelet aggregation of thromboxane A2 and, thus, prevents thrombus formation [31]. Similar to ASA, extract S6, but interestingly not extract B, could block both COX-1 and COX-2 enzyme activities, while a comparable COX-1 inhibition was observed at a five-times lower concentration compared to COX-2. This indicates essential bioactivity differences between the *Salix* species. The extracts used in this study were standardized to their total phenolics content.

A major difference between the two plant extracts was the presence of 2′-*O*-acetylsalicortin; however, this had no effect on both COX-1 and COX-2 enzyme activities. Thus, so far, we cannot explain this difference, but it deserves more attention in future work. Interestingly, we observed that *Salix* extracts and ASA did not inhibit COX-2 protein expression in activated PBMCs. This contradicts another study, which reported that the aqueous willow bark extract STW 33-I (with 23–26% salicin) and also ASA exhibited a significant inhibitory effect on COX-2 mRNA expression in IFN-γ/LPS-activated monocytes [32]. This discrepancy could be due to different inflammatory stimulators but also the duration of extract exposure (6 h vs. 48 h of stimulation) in addition to the compound composition.

Even though NSAIDs are already commonly used to relieve pain and bring down fever in SARS-CoV-2 infected patients, there is still controversy on whether they could be harmful to patients. Early in the COVID-19 pandemic, several hypotheses were generated on whether NSAIDs could upregulate angiotensin-converting enzyme 2 (ACE-2), the essential SARS-CoV-2 viral cell entry receptor, and subsequently result in an increased risk of SARS-CoV-2 infection [33,34]. According to these concerns, a report from French authorities raised a warning against NSAIDs usage, particularly for ibuprofen, in COVID-19 patients [35].

At this time, the World Health Organization (WHO) and other public health officials from UK and Italy also suggested not using ibuprofen in patients with COVID-19 symptoms. However, they later reversed their recommendation because there was no real evidence that using NSAIDs, especially ibuprofen, made the infection worse [36]. The European Medicines Agency (EMA) also disagreed with the French report [37]. According to a new study from 2021, the inhibition of PG production by ibuprofen and meloxicam could ultimately influence COVID-19 outcomes by dampening inflammatory cytokine and antibody responses in SARS-CoV-2-infected mice.

Additionally, they found that the inhibition of COX-2/PGE_2_ signaling by those NSAIDs had no impact on the SARS-CoV-2 virus entry receptor ACE-2 expression, viral entry, or viral replication [7]. Furthermore, in April 2021, another NSAID, namely naproxen, was demonstrated to have direct antiviral activity on SARS-CoV-2 replication by the inhibition of nucleoproteins oligomerization of SARS-CoV-2. The success of those in vitro results currently led them to investigate the effect of naproxen in a clinical trial with severely ill patients [38].

The excessive production of pro-inflammatory IL-6 has been reported as predictive of poor outcomes during COVID-19 infection and was highly associated with mortality [39,40]. In our in vitro study, we observed that *Salix* extract S6 could block IL-6 production with a much greater potency as compared to ASA. However, a strong elevation of pro-inflammatory TNF-α and decrease of IL-10, which is considered anti-inflammatory, was also seen in SARS-CoV-2 peptide/IL-1β stimulated PBMCs upon *Salix* extract S6 exposure.

An increase in TNF-α production in rheumatoid synovia and LPS-stimulated human monocytes was reported after treatment with different NSAIDs (i.e., celecoxib, rofecoxib, and diclofenac) [41]. Other NSAIDs (i.e., diclofenac sodium, indomethacin, and nabumethon) raised the TNF-α serum level in osteoarthritis patients [42]. Both studies explained that decreased levels of PGE_2_ caused by NSAIDs could possibly increase the expression of key pro-inflammatory cytokine TNF-α, which could be due to a negative feedback loop from PGE_2_ [43].

Even though IL-10 is considered anti-inflammatory, Lu et al. hypothesized that it could also play a pro-inflammatory and immune-activating role in COVID-19 pathogenesis. They proposed that increased endogenous IL-10 production during the initiation phase of SARS-CoV-2 infection might function as an immune activating/pro-inflammatory agent that subsequently stimulated the secretion of other inflammatory mediators of the cytokine storm [20]. Thus, we do not currently know whether these in vitro effects of *Salix* extracts on relevant cytokines can be regarded as adverse or beneficial effects in the context of COVID-19.

In addition to fever and respiratory symptoms, gastrointestinal (GI) symptoms, such as anorexia, diarrhea, vomiting, and abdominal pain, have been massively demonstrated as a distinctive feature of COVID-19 [44]. A meta-analysis from 125 articles with a total of 25,252 patients showed that 20.3% COVID-19 patients experienced GI manifestations [45]. This manifestation could be explained by the abundant expression of the main entry receptor for SARS-CoV-2, ACE-2, in the digestive tract. Once the virus enters the epithelium and directly damages the intestinal cells, this subsequently results in injury and inflammation of the GI epithelium [46].

Using an in vitro model of small intestinal inflammation based on a human Caco-2/HT29-MTX cell line co-culture, we found that *Salix* extracts exhibited its anti-inflammatory effects by inhibiting PGE_2_ release here, as well. In 2021, Crittenden et al. reported the PGE_2_-mediated promotion of intestinal inflammation via inhibiting microbiota-dependent regulatory T-cells. However, thus far, there are no other reports on PGE_2_ regulation in the context of SARS-CoV-2-mediated intestinal inflammation [47].

Furthermore, there is in vitro evidence for sustained anti-inflammatory activity of *Salix* extracts in activated PBMCs after intestinal absorption and hepatic metabolism using the hepatocyte-like HepaRG cell line [48] in addition to the intestinal Caco-2/HT29-MTX co-culture. This is an advancement based on the most common in vitro co-culture models involving the human intestinal Caco-2 and hepatic HepG2 cells [49,50,51]. Compared to the monoculture of differentiated Caco-2 cells, which is a well-established model for permeability to predict oral compound absorption in humans [52,53], the additional presence of mucus producing goblet HT29-MTX cells in this co-culture system could help to avoid an in vitro overestimation of compound permeability, as described earlier [54,55,56].

This is due to the fact that mucus production acts as interactive barrier to limit the free diffusion of small compounds from the apical cell surface [57]. In addition, various cytotoxicity and toxicogenomic studies have demonstrated that HepaRG cells present a more accurate human hepatocyte-like model than other hepatic cell lines, including HepG2 cells, because differentiated HepaRG cells highly maintain liver-specific functions, such as drug-metabolizing enzymes and activities [58,59,60].

NSAIDs involving ASA are already in off-label use by many clinicians for the treatment of viral infections, including COVID-19. Since robust evidence of safety or effectiveness is still missing, it is still a controversial topic whether NSAIDs are beneficial or harmful to SARS-CoV-2 infected patients. The timing of treatment may present a critical factor here and this might also be true for herbal medicines. There is an arising voice that herbal medicines, which have been used to prevent respiratory viral infection for years, currently present promising candidates as adjuvants during COVID-19 [61].

We demonstrated here the potential therapeutic efficacy of *Salix* extracts, which were comparable or even superior to ASA in vitro upon SARS-CoV-2 peptide/IL-1β challenge. Studies on osteoarthritis and rheumatoid arthritis [62] have reported that adverse effects from *Salix* formulations were minimal as compared to NSAIDs, including ASA, and thus allergic reactions in salicylate-sensitive individuals might be the primary cause for concern here.

Despite its many advantages, precautions should be taken related to the limitations of the human PBMC model. First, we used PBMCs from healthy donors and not from recovered or infected SARS-CoV-2 patients. Moreover, stimulation was done here with a SARS-CoV-2 peptide pool, not the virus itself. Infection and inflammation surely influence the reactivity of immune cells and the response to *Salix* treatment here might be different from what can be observed upon ex vivo stimulation of cells from healthy donors with SARS-CoV-2 virus or peptide pool. Therefore, further in vitro and in vivo studies using *Salix* extracts upon SARS-CoV-2 infection are now necessary to gain a deeper mechanistic understanding and confirmation.

## 4. Materials and Methods

### 4.1. Chemicals

Fetal calf serum (FCS), L-glutamine solution, RPMI-1640, DMEM, William’s Medium E + GlutaMAX™, human recombinant insulin zinc solution (4 mg/mL), trypsin-EDTA 1000 × (5 and 2.2 mg/mL), trypsin (0.5%) solution and phosphate buffered saline (PBS, without Ca^2+^ and Mg^2+^), Non-Essential Amino Acid (NEAA), and penicillin/streptomycin solution (10,000 U/mL and 10,000 µg/mL) were purchased from Gibco™, Life Technologies GmbH (Darmstadt, Germany).

Lipopolysaccharide (LPS, from *Escherichia coli* O11:B4), phorbol 12-myristate 13-acetate, hydrocortisone 21-hemisuccinate sodium salt, 2,2-diphenyl-1-picrylhydrazyl, ethanol absolute, ascorbic acid, 2,4,6-tris(2-pyridyl)-s-triazine, and ferric chloride hexahydrated were from Sigma Aldrich (Taufkirchen, Germany). LymphoPrep^TM^ gradient was purchased from Progen (Heidelberg, Germany). ROTISOLV^®^ HPLC gradient grade and Triton-X100 were purchased from Carl Roth (Karlsruhe, Germany). Dimethyl sulfoxide (DMSO; purity > 99%) was purchased from Applichem GmbH (Darmstadt, Germany). Sodium acetate, ferrous sulphate heptahydrate, acetic acid, and hydrochloric acid (1 mol/L) were purchased from Merck KGaA (Darmstadt, Germany).

Anti-human COX-2 was from R&D System (Wiesbaden, Germany). mAb against β-actin was from Sigma Aldrich (Taufkirchen, Germany). The horseradish peroxidase (HRP)-labeled secondary antibodies, anti-mouse and anti-goat were from Cell Signaling Technology (Boston, MA, USA).

### 4.2. Extracts and Standard Preparation

One-year-old branches of the clone PE1 from willow species *Salix pentandra* plants (extract S6) were cut off in August 2016. The bark was peeled, frozen (−80 °C) and immediately lyophilized. For the extraction of phenolics with the aim of quantification, 20 mg pulverized bark material was extracted with 500 µL 70% methanol (containing 0.1% formic acid) in an ultrasonic bath of ice water for 15 min. After centrifugation (9000× *g*, 5 min, and room temperature), the supernatant was collected, and the pellet was re-extracted with 200 µL of the extraction solution, twice. The combined supernatants were concentrated in a vacuum concentrator to near dryness.

To determine the phenolic content in the concentrated extracts, a slightly modified method was used [63]. Thereby, the extracts (using duplicates for each extract) were refilled with 100 µL internal standard (resorcinol, 50 mM) and ultrapure water up to 1 mL. The extract was filtered using SpinX tubes (0.22 µm) and stored at −20 °C until HPLC analysis. The HPLC system consisted of a DIONEX P680 pump, an ASI-100 auto sampler, a TCC-100 thermally-regulated column department, and an UltiMate 3000 Photodiode Array Detector. The software Chromeleon 6.8 was used for peak evaluation.

Reversed phase chromatography was carried out on an Acclaim PolarAdvantage C16 column (3 μm, 120 Å, 2.1 × 150 mm, Thermo-Fisher, Scientific GmbH, Dreieich, Germany) protected by a pre-column (5 μm, 120 Å, 2 × 10 mm, Thermo-Fisher, Scientific GmbH, Dreieich, Germany). The eluents used for HPLC analysis were (A) 2% tetrahydrofuran, 0.5% phosphoric acid in ultrapure water, and (B) 100% methanol. The extracts were analyzed with a flow rate of 0.35 mL/min and the following gradient program: 0% B (0–5 min), 0–15% B (5–10 min), 15–25% B (10–20 min), 25–35% B (20–30 min), 35–50% B (30–40 min), 100% B (40–42 min), 100–0% B (42–44 min), and 0% (44–49 min). The injection volume was 10 µL, and peak detection was carried out at 270 nm.

Qualitative analysis of the phenolics was based on their retention times, specific UV-spectra [64], and mass spectrometry; quantitative analysis was based on the peak area in relation to the internal standard. To correct for absorbance differences, response factors were used [65]. The phenolic content was calculated in mg/mL dry weight (DW). To prepare standardized extracts of *Salix* with a phenolic content of 10 mg/mL, the concentrated extract was diluted in an adapted amount of distilled water and then filtered. All work steps for preparing the standards were performed in sterile conditions and with sterile materials.

A willow bark reference (extract B), which is used for phytopharmaceutical production, was provided from Bionorica SE (Neumark, Germany). A total of 20 mg of dry extract was filled with 100 µL internal standard (resorcinol, 50 mM) and ultrapure water up to 1 mL, dissolved in an ultrasonic bath for 15 min, and filtered using SpinX tubes (0.22 µm). Qualitative and quantitative analysis of phenolics was done in duplicate using the same method as described for the *Salix* extracts above. The extracts were standardized to 10 mg/mL phenolic content.

### 4.3. Isolation of 2′-O-Acetylsalicortin from Extract S6

To gain deeper insight into the inhibitory COX1/2 enzyme activity of extract S6, the most abundant compound, 2′-*O*-acetylsalicortin, was isolated and purified by (semi-)preparative HPLC analysis. Therefore, ground willow bark powder (120 g) was extracted sequentially four times with methanol, three times with methanol/water (*v*/*v*, 70/30), and another three times with water. The lyophilized methanol extract containing high amounts of 2′-*O*-acetylsalicortin was used for solid-phase extraction (SPE). Elution of the SPE fraction F1 was performed by use of C18 cartridges (CHROMABOND^®^, Macherey-Nagel GmbH & Co. KG, Düren, Germany) applied with 10 g methanol extract in 70 mL distilled water.

Another ten fractions were eluted in 10%-steps using methanol/water mixtures with increasing methanol content as solvents (SPE fraction F2 eluted with *v*/*v* 10/90 methanol/water). The freeze-dried SPE fraction F5 (eluted with *v*/*v* 60/40 methanol/water) was purified further and pure 2′-*O*-acetylsalicortin was isolated by means of UV-HPLC using a preparative 250 × 21.2 mm, 5 µm, Luna^®^ phenyl-hexyl and semi-preparative 250 × 10 mm, 5 µm, Luna^®^ pentafluorophenyl (PFP) column.

The following settings were applied for preparative pre-fractionation: UV detection at 200 nm and a 20 mL/min flow rate. We used 0.1% formic acid in water (solvent A) and 0.1% formic acid in acetonitrile (solvent B) as solvents, and the following gradient was applied: starting with a mixture of 22% B, held for 3 min, increased in 10 min to 23.5% B, continued for 15 min at isocratic flow, decreased again to the initial conditions of 22% B, and held for 3 min. Further, 2′-*O*-acetylsalicortin was purified by semi-preparative sub-fractionation using 0.1% formic acid in water (solvent A), 0.1% formic acid in methanol (solvent B), and UV detection at a wavelength of 200 nm.

The chromatographic conditions at a flow rate of 4.7 mL/min started at 22% solvent B, held for 3 min, increased in 15 min to 27.7% B, held for 2 min, increased to 52% B in 10 min, isocratic for 3 min, then increased further in 20 min to 57% B, held for 2 min, finally decreased in 3 min to 22% B, and isocratic for 2 min.

Structure elucidation was performed by means of one- and two-dimensional NMR analysis on a AVANCE III 500 MHz spectrometer (Bruker, Rheinstetten, Germany) as well as high-resolution mass spectrometry on a Synapt G2 HDMS UPLC-ToF-MS system (Waters UK Ltd., Manchester, UK).

### 4.4. Isolation and Cultivation of Human PBMCs

The study was approved by the ethics committee of the University of Freiburg. Human PBMCs were isolated from fresh peripheral blood or buffy coats of healthy volunteers at the University Medical Center in Freiburg, Germany. Blood was collected using Li-heparinized vacutainers (Sarstedt, Nümbrecht, Germany) after obtaining written informed consent of the volunteers. PBMCs were isolated from blood by centrifugation on a LymphoPrepTM gradient (density: 1.077 g/cm^3^, 20 min, 500× *g*). Isolated PBMCs were cultured in RPMI 1640 medium supplemented with 10% heat-inactivated FCS, 2mM L-glutamine, 100 U/mL penicillin, and 100 µg/mL streptomycin at 37 °C in a humidified incubator with a 5% CO_2_/95% air atmosphere.

### 4.5. Stimulation of Isolated PBMCs with SARS-CoV-2 Overlapping Peptide Pools or LPS

PepTivator^®^ SARS-CoV-2 Prot_S Peptide Pools purchased from Miltenyi Biotec (Cologne, Germany) or bacterial LPS were used for PBMCs stimulation. The overlapping peptide pools (OPP), which consist of 15-mer sequences with an 11 amino acid overlap, represent the immunodominant sequence domains of the surface glycoprotein of SARS-CoV-2, the so-called “spike” protein. As described before [26], after pre-treatment with *Salix* extracts or ASA for 30 min, 1 × 10^6^ isolated PBMCs were either co-stimulated with 1 µg/mL OPP and 50 ng/mL IL-1β or 100 ng/mL LPS for different time points in a humidified incubator with 5% CO_2_/95% air atmosphere at 37 °C. The negative control was left untreated.

### 4.6. Cell Culture

The human colon carcinoma Caco-2 (ACC169) and HT29-MTX-E12 (from now, called HT29-MTX) cell lines were obtained from the German Collection of Microorganisms and Cell Cultures (DSMZ, Braunschweig, Germany) and European Collection of Authenticated Cell Cultures (Porton Down, UK), respectively. The cells were cultured separately in flasks in Dulbecco’s Modified Eagle’s medium (DMEM) supplemented with 10% FCS, 1% Non-Essential Amino Acid (NEAA), 100 U/mL penicillin, and 100 µg/mL streptomycin at 37 °C in a humidified incubator with a 5% CO_2_/95% air atmosphere. The culture medium was changed every 2–3 days.

The human hepatic cell line HepaRG was obtained from Biopredic International^®^ (Rennes, France). The cell line was cultured in William’s Medium E + GlutaMAX™, supplemented with 10% of FCS, 100 U/mL penicillin, and 100 µg/mL streptomycin, 50 µM hydrocortisone 21-hemisuccinate sodium salt, and 5 μg/mL human insulin. The maintenance and differentiation into fully functional hepatocytes-like cells was performed according to Biopredic International^®^ instructions. Briefly, cells were grown for 14 days in culture medium. After this stage, cells were differentiated for another 14 days with the addition of DMSO to the culture medium.

The first 3 days of differentiation with 1% DMSO (*v*/*v*) in the culture medium, and the following days with 2% DMSO (*v*/*v*). After the differentiation stage, the cells were kept in culture medium with 0.5% DMSO (*v*/*v*) for seven days and then used in the experiments. During the culture, the cells were visualized by microscope regularly and after differentiation, and granular hepatocyte-like cells in which many bright canaliculi-like structures were recognized. The cells were maintained at 37 °C in a humidified incubator with a 5% CO_2_/95% air atmosphere.

### 4.7. Cell Co-Culture and Corresponding Treatments

For modelling gut inflammation, monocultures of Caco-2 and HT29-MTX cells were harvested with Trypsin-EDTA and seeded on the apical chamber of 12-well ThinCert^®^ inserts (0.4 µm PET pore membrane, Greiner Bio-One, Frickenhausen, Germany) in an optimal proportion of 90:10 as described before [56,57], respectively, to reach a final density of 3 × 10^5^ cells/cm^2^ in each insert. Cells were co-cultured for 19–21 days in a humidified incubator with a 5% CO_2_/95% air atmosphere with medium (0.5 mL in the apical side and 1.5 mL in the basolateral side) changed every 2–3 days. After pre-treatment with *Salix* extracts or ASA for 30 min, the cells were exposed with inflammatory co-stimulators, including IL-1β (25 ng/mL), IFN-γ (50 ng/mL), TNF-α (50 ng/mL), and LPS (1 µg/mL) for 24 h in a humidified incubator with a 5% CO_2_/95% air atmosphere at 37 °C. The negative control was left untreated.

For intestinal absorption and liver metabolism experiments, a complex co-culture system was used. For this, the Caco-2/HT29-MTX co-culture after differentiation on the inserts (i.e., apical side) was incubated together with differentiated HepaRG cells on a 12-well culture plate (i.e., basolateral side). *Salix* extracts or ASA were put into the apical side for 4 h before the supernatant from the basolateral side was collected. This supernatant was immediately used for incubation with freshly isolated PBMCs 30 min before treatment with 100 ng/mL LPS for another 24 h.

### 4.8. Quantification of PGE_2_ Release Using ELISA Technique

Cells were stimulated as described above, and cell-free supernatants were collected and frozen at −80 °C until analysis for PGE_2_ release using the PGE_2_ ELISA kit from Cayman (Hamburg, Germany) according to the manufacturer’s instructions.

### 4.9. Cytokine Determination Using ELISA Technique

Cells were stimulated as described above, and cytokine (IL-1β, IL-6, and IL-10) secretions were evaluated in the supernatants using ELISA kits (Thermo Scientific, Darmstadt, Germany) according to the manufacturer’s instructions.

### 4.10. Inhibition of (Human Recombinant) COX1/2 Enzyme Activity

The inhibition of extract on COX enzyme activity was quantified with the Cayman COX (human) Inhibitor Screening Assay kit (Cayman, Hamburg, Germany) according to the manufacturer’s instructions. Briefly, the test extract or compound was incubated with the mixture of reaction buffer solution, heme, and COX enzyme in 8 min at 37 °C. After that, this solution was incubated for exactly 30 s with Arachidonic Acid (AA) followed by stopping the COX reaction with the saturated Stannous Chloride. The next step was the quantification of the prostaglandins PGF_2α_ released from the solution via ELISA assay.

### 4.11. Intestinal Cell Monolayer Integrity

The cell monolayer integrity of the Caco-2/HT29-MTX co-culture was checked on days 19–21 of culture using transepithelial electrical resistance (TEER) measurements performed with an EVOM epithelial volt-ohmmeter equipped with ‘chopstick’ electrodes (Millicell^®^ ERS, Millipore, Bedford, MA, USA) [66]. Co-culture inserts with TEER values over 200 Ω·cm^2^ were used for further experiments.

### 4.12. Protein Analysis Using Immunoblotting

The analysis of COX-2 protein expression was performed in human PBMCs either with or without stimulation with 1 µg/mL SARS-CoV-2 peptide mixture and 50 ng/mL IL-1β, or 100 ng/mL LPS. Cell lysis and immunoblotting was conducted as reported previously [66]. *β*-actin was used as the loading control.

### 4.13. Statistical Analysis

Data are the means ± SD of at least three independent experiments. In studies with PBMCs, blood from a different donor was used for each experiment. When comparing multiple means, the results were analyzed either by one-way ANOVA or one-way ANOVA followed by Dunnett’s multiple comparison tests.

## Figures and Tables

**Figure 1 ijms-22-06766-f001:**
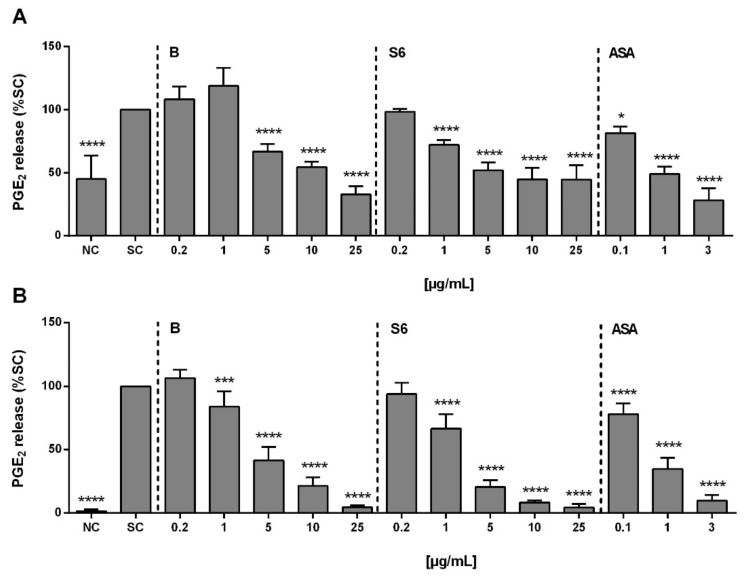
The effects of *Salix* extracts and acetylsalicylic acid (ASA) on the inhibition of PGE_2_ release from activated peripheral blood mononuclear cells (PBMCs). Cells (1 × 10^6^ cells/mL) were pre-treated with *Salix* extracts B or S6 or ASA 30 min before stimulation for another 24 h with either (**A**) SARS-CoV-2 peptide mixture (1 µg/mL) and IL-1β (50 ng/mL) or (**B**) LPS (100 ng/mL). Negative control (NC): cells only. Solvent control (SC): 1% distilled water in activated cells. The amount of PGE_2_ in the culture supernatant was measured using an ELISA kit. Bars are the means ± standard deviation (SD) (*n* ≥ 3). * *p* < 0.05, *** *p* < 0.001, or **** *p* < 0.0001 versus the SC group.

**Figure 2 ijms-22-06766-f002:**
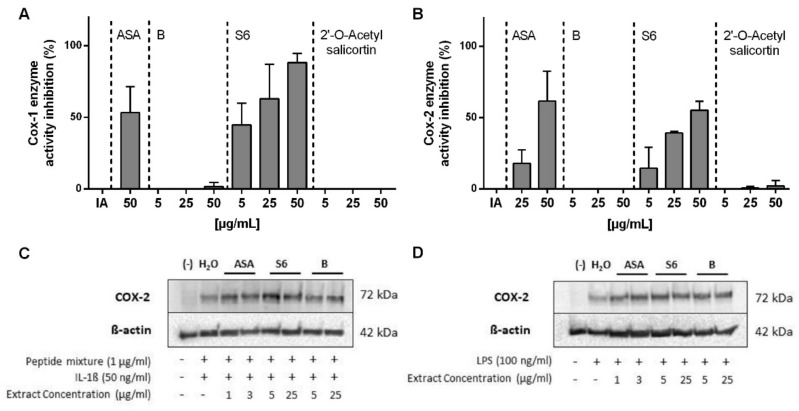
The effects of *Salix* extracts and ASA on COX1/2 enzyme activity. Human recombinant (**A**) COX-1 or (**B**) COX-2 enzyme activity was quantified by detection of prostaglandin PGF_2α_ production using the ELISA technique. The inhibition (%) was calculated by comparison to the Initial Activity (IA) of the respective COX enzyme. Bars are the means ± SD (*n* ≥ 3). Western blot of COX-2 protein expression. PBMCs (1 × 10^6^ cells/mL) were pre-treated with *Salix* extracts or ASA for 30 min and subsequently stimulated with (**C**) 1 µg/mL peptide mixture and 50 ng/mL IL-1β or (**D**) 100 ng/mL LPS for another 3 h. Cells were lysed, and the total lysate was subjected to western blotting. Representative immunoblots of COX-2 and *β*-actin (loading control) are shown.

**Figure 3 ijms-22-06766-f003:**
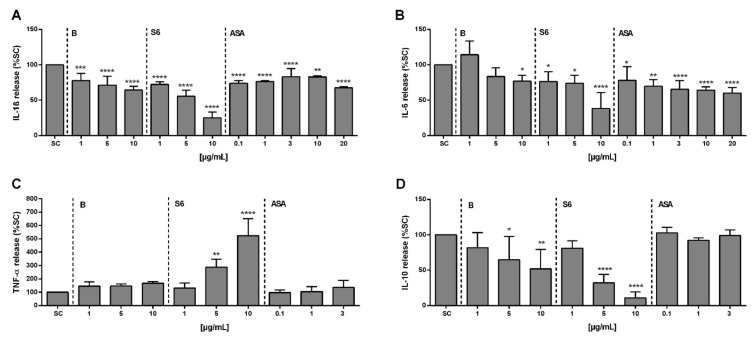
The effects of *Salix* extracts and ASA on the inhibition of cytokine secretion from SARS-CoV-2 peptide/IL-1β-stimulated PBMCs. Cells (1 × 10^6^ cells/mL) were pre-treated with *Salix* extracts or ASA for 30 min before stimulation for 6 h (TNF-α detection) or 24 h (interleukins). The amount of (**A**) IL-1β, (**B**) IL-6, (**C**) TNF-α, and (**D**) IL-10 in the cell supernatant was measured using the ELISA technique. SC: 1% distilled water in activated cells. Bars are the means ± SD (*n* ≥ 3). * *p* < 0.05, ** *p* < 0.01, *** *p* < 0.001, or **** *p* < 0.0001 versus SC group.

**Figure 4 ijms-22-06766-f004:**
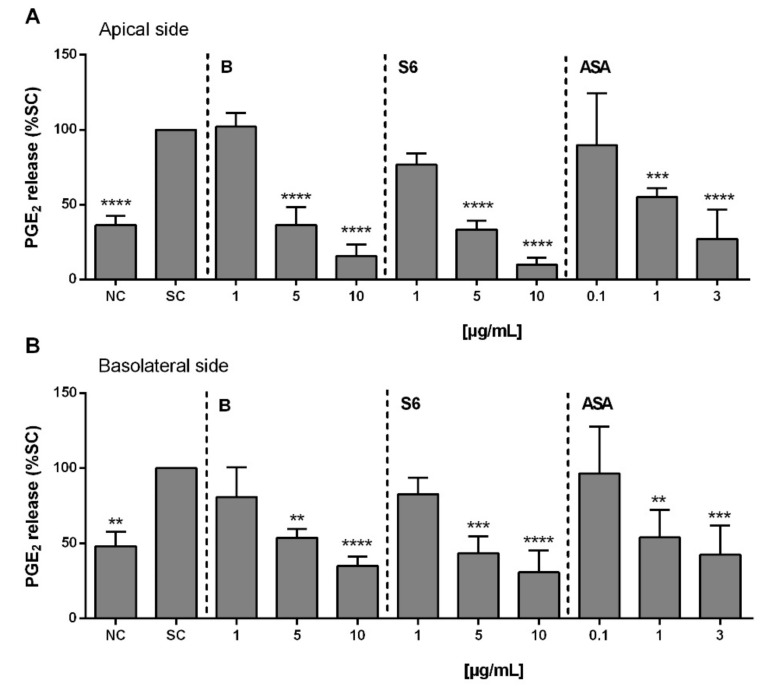
The effects of *Salix* extracts and ASA on the inhibition of PGE_2_ release from activated Caco2/HT29-MTX co-culture. After 19–21 d of co-culture, the cells were pre-treated with *Salix* extracts or ASA for 30 min before activation for 24 h with IL-1β (25 ng/mL), IFN-γ (50 ng/mL), TNF-α (50 ng/mL), and LPS (1 µg/mL). The amount of PGE_2_ in supernatant of the (**A**) apical side and (**B**) basolateral sides was quantified using an ELISA kit. NC: cells only. SC: 1% distilled water in activated cells. Bars are the means ± SD (*n* ≥ 3). ** *p* < 0.01, *** *p* < 0.001, or **** *p* < 0.0001 versus the SC group.

**Figure 5 ijms-22-06766-f005:**
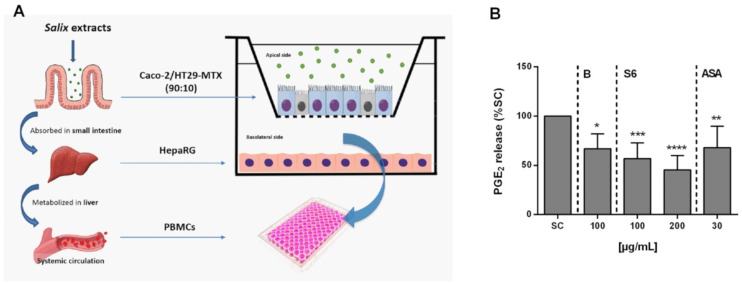
The anti-inflammatory effect of *Salix* extracts and ASA in LPS-stimulated PBMCs after intestinal absorption and liver cell metabolism. (**A**) Schematic diagram of the in vitro triple co-culture model system. (**B**) Inhibition of PGE_2_ secretion from LPS-stimulated PBMCs after in vitro absorption and metabolism. After 4 h incubation of *Salix* extracts and ASA in the co-culture model system, the supernatant from the basolateral side was collected and immediately incubated with PBMCs (1 × 10^6^ cells/mL) for 30 min before LPS (100 ng/mL) stimulation for 24 h to determine the remaining anti-inflammatory capacity. PGE_2_ was quantified using an ELISA kit. SC: 1% distilled water in activated cells. Bars are the means ± SD (*n* ≥ 3). * *p* < 0.05, ** *p* < 0.01, *** *p* < 0.001, or **** *p* < 0.0001 versus the SC group.

**Table 1 ijms-22-06766-t001:** Phenolic compounds quantified in 10 mg/mL *Salix* extracts.

Chemical Group	Phenolic Compound	Extracts [mg/mL]	
		B	S6
Salicylates	Salicin	2.26 ± 0.03	0.08 ± 0.00
	Salicortin	2.79 ± 0.04	-
	2′-*O*-acetylsalicin	-	0.76 ± 0.01
	2′-*O*-acetylsalicortin	-	7.31 ± 0.01
	Tremulacin	0.30 ± 0.00	-
Flavonoid *O*-glycosides	Naringenin-5-glucoside I	1.26 ± 0.01	-
	Naringenin-5-glucoside II	1.63 ± 0.01	-
	Naringenin-7-glucoside	0.52 ± 0.00	-
	Luteolin-7-glucoside	0.21 ± 0.00	-
	Quercetin-hexoside	0.06 ± 0.00	0.19 ± 0.00
	Isosalipurposide	0.10 ± 0.00	-
Flavonoid aglycones	Catechin	-	0.78 ± 0.00
	Epicatechin	-	0.03 ± 0.00
Others	Triandrin	0.10 ± 0.03	-
	Caffeic acid derivative I	0.14 ± 0.03	-
	Caffeic acid derivative II	0.17 ± 0.00	-
	Syrengin	0.46 ± 0.01	0.85 ± 0.01

## Data Availability

Not applicable.

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
