# Peer review of "Comparative Anti-Inflammatory Effects of *Salix* Cortex Extracts and Acetylsalicylic Acid in SARS-CoV-2 Peptide and LPS-Activated Human In Vitro Systems"

_ijms, 2021, doi:10.3390/ijms22136766_

Round 1
Reviewer 1 Report
The investigation of anti-inflammatory drugs for COVID-19 therapy is the actual goal of the molecular biology and medicine. Authors investigate anti-inflammatory properties of Salix extracts in cell cultures. Results, obtained in this investigation, are important. This investigation has a nice design and modern methodology.
However, we have some questions and comments for authors.
- It is recommended no mention drugs commercial name «Aspirin®» in the title of manuscript. The manuscript title can be formulated as «Comparative anti-inflammatory effects of Salix cortex extracts 2 and acetylsalicylic acid in SARS-CoV-2 peptide and 3 LPS-activated human in vitro systems». Also it is recommended to change in all manuscript text the commercial name «Aspirin®» on «acetylsalicylic acid».
- Line 89 «PGE2 concentration was 177.6±111.2 ng/ml». There SD is more than 30% of mean value. Is it the error?
- Line 428-429 «After pre-treatment with Salix extracts or Aspirin® for 30 428 min… ». Why authors choose time interval 30 min? If it was the literature data, it needs to add the reference. If this time was chosen in previously experiments, it needs to write about this.
- Line 444-445 «The maintenance 444 and differentiation into fully functional hepatocytes-like cells was performed according 445 to Biopredic International® instructions». Could authors shortly describe the method for verification of cell differentiation? If it was described previously, could authors give the reference?
- It was used primary human vessels (PBMCs), human carcinoma (line CaCo-2, line HT29-MTX) and human hepatic cell line HepaRG. Cancer cells have different metabolism and different signaling answers on drugs in some causes. Why in this work cancer cell lines are the equate model for investigation of anti-inflammatory properties of Salix extracts? Primarily lung cells involved in COVID-19 pathogenesis. Why authors didn’t do experiments in primary/line of human lungs cells?
Reviewer 2 Report
- In Figure's 1 legend the authors need to explain what B and S6 are.
- Page 1, line89-90 "Upon activation, the PGE2 concentration in PBMCs was 177.6 ± 111.2 ng/ml, which was 155% compared to untreated control cells". Is this the SC sample? Is the SC stimulated PBMCs without extract or ASA? This needs to be clearly stated in the manuscript.
- In the SC samples in Figure 1 did you notice any difference in PGE expression in PBMCs stimulated with the SARS-cov-2 peptide pool vs the LPS?
- In Figure 1, why the NC in panel B is so different than in panel A since it is just unstimulated PBMCs in both cases?
- How many donors did you use for your experiments? (i.e. you isolated PBMCs from how many healthy donors)
- In section 2.3 and Figure 3, what are the levels of each cytokine in non-activated cells (NC)? Did you notice elevated levels upon stimulation of PBMCs with the peptide pool?This needs to be provided/shown in the manuscript.
- In Figure 4 the levels of non-activated cells (NC) are not shown. This needs to be shown in the Figure.
- In the Discussion, lines 244-245 "Besides, COXs had also been described to regulate the replication of mouse hepatitis coronavirus (MHV) before". I don't see how this sentence and the context fit in that part of the text. Please rephrase and/or elaborate.
-
In the Discussion, line 319 "However, we also found differences in the mechanism of action in terms of COX-2 enzyme activity inhibition". Why does the sentence start with "However"? Also, what are these differences as it is not clear. Please elaborate.
- In the results section when the authors start presenting the experiments with the PBMCs they need to clarify that they are obtained from healthy volunteers.
- The authors need to mention at least in the Discussion the limitations of their study. The most important one is the fact that the experiments have been performed with PBMCs from healthy individuals and not from Covid-19 recovered donors or patients. This means that even though the authors "activate" the PBMCs with the peptide pool, the cells have never encountered the pathogen. So the immune response only relies in pre-existing immunity probably from circulating coronaviruses that cause the common cold as it has been evidenced. The authors need to be careful with phrases like "In this study, we aimed at evaluating the in vitro anti-inflammatory efficiency of Salix 71 cortex extracts in comparison to Aspirin® in the context of SARS-CoV-2 infection" in the introduction or "in the context of COVID19" in the abstract. In this study since only healthy volunteers have been used it is not proper to translate the results in context of COVID 19 disease. Needs to be clarified that the results are based on the SARS-cov-2 peptide pool and not the virus.
Round 2
Reviewer 2 Report
Most of the comments have been carefully addressed and the article should be published. Overall, this is a clear, concise, and well-written manuscript.